

# Do triceps surae muscle dynamics govern non-uniform Achilles tendon deformations?

William H. Clark and Jason R. Franz

Joint Department of Biomedical Engineering, University of North Carolina at Chapel Hill and North Carolina State University, Chapel Hill, NC, United States of America

## ABSTRACT

The human Achilles tendon (AT) consists of sub-tendons arising from the gastrocnemius and soleus muscles that exhibit non-uniform tissue displacements thought to facilitate some independent actuation. However, the mechanisms governing non-uniform displacement patterns within the AT, and their relevance to triceps surae muscle contractile dynamics, have remained elusive. We used a dual-probe ultrasound imaging approach to investigate triceps surae muscle dynamics (i.e., medial gastrocnemius-GAS, soleus-SOL) as a determinant of non-uniform tendon tissue displacements in the human AT. We hypothesized that superficial versus deep differences in AT tissue displacements would be accompanied by and correlate with anatomically consistent differences in GAS versus SOL muscle shortening. Nine subjects performed ramped maximum voluntary isometric contractions at each of five ankle joint angles spanning $10°$ dorsiflexion to $30°$ plantarflexion. For all conditions, SOL shortened by an average of 78% more than GAS during moment generation. This was accompanied by, on average, 51% more displacement in the deep versus superficial region of the AT. The magnitude of GAS and SOL muscle shortening positively correlated with displacement in their associated sub-tendons within the AT. Moreover, and as hypothesized, superficial versus deep differences in sub-tendon tissue displacements positively correlated with anatomically consistent differences in GAS versus SOL muscle shortening. We present the first *in vivo* evidence that triceps surae muscle dynamics may precipitate non-uniform displacement patterns in the architecturally complex AT.

## INTRODUCTION

The Achilles tendon (AT) is a critical passive elastic structure that transmits contractile forces from the gastrocnemius and soleus muscles (i.e., triceps surae) to generate a moment about the ankle, thereby powering activities such as walking (*Fukashiro, Hay & Nagano, 2006*). In young adults, the power generated via triceps surae muscle–tendon interaction during walking is responsible for 70%–80% of the mechanical power needed for forward propulsion and swing initiation (*Neptune, Clark & Kautz, 2009*; *Whittington et al., 2008*; *Zelik et al., 2014*). The architecturally complex AT consists of three distinct bundles of tendon fascicles, known as "sub-tendons" (*Handsfield et al., 2017*), arising from the medial and lateral gastrocnemius and soleus muscles (*Bojsen-Moller & Magnusson, 2015*;

Corresponding author
Jason R. Franz, jrfranz@email.unc.edu

*Edama et al., 2015*; *Szaro et al., 2009*). Recent advances in the use of dynamic, *in vivo* ultrasound imaging have revealed non-uniform displacement patterns within the AT. Those non-uniform patterns are commonly interpreted as evidence for sliding between adjacent sub-tendons, which has the potential to facilitate independence between the individual triceps surae muscles (*Arndt et al., 2012*; *Chernak, Slane & Thelen, 2014*; *Franz & Thelen, 2015*). However, the mechanisms governing these non-uniform displacement patterns, and their relevance to triceps surae muscle contractile dynamics, remain elusive.

To our knowledge, no empirical studies have characterized the origins of sliding between adjacent sub-tendons within the human AT. However, recent model predictions implicate differential gastrocnemius versus soleus muscle dynamics as a plausible candidate (*Handsfield et al., 2017*). Using anatomy derived from MRI, *Handsfield et al. (2017)* tuned a finite-element model of the AT to replicate published estimates of *in vivo* AT tissue displacements during isolated contractions (*Chernak, Slane & Thelen, 2014*). The authors interrogated their model to reveal that without differential muscle forces, AT tissue non-uniformity decreased by 85% (*Handsfield et al., 2017*). One implication of this prediction is that triceps surae muscle dynamics may precipitate non-uniform displacement patterns in the human AT –a finding for which experimental evidence has not yet been presented. Indeed, other factors, including differences in sub-tendon material or architectural properties or in calcaneal insertion have also been implicated (*Franz & Thelen, 2015*; *Thorpe et al., 2012*; *Thorpe et al., 2013*).

Cine B-mode imaging has revealed important disparities between the individual triceps surae muscles. For example, *Ishikawa et al. (2005a)* found that gastrocnemius muscle fascicles remain isometric or shorten during the late stance phase of walking while soleus muscle fascicles lengthen (*Ishikawa, Niemela & Komi, 2005b*). *Cronin et al. (2013)* added that the contractile behavior of the gastrocnemius, but not the soleus, changed with walking speed (*Cronin et al., 2013*). These data allude to the potential for functional benefits at the muscle level conferred by independent actuation –actuation that may be facilitated by the AT itself (*Huijing et al., 2011*; *Kawakami, Ichinose & Fukunaga, 1998*; *Tian et al., 2012*). However, we recently observed cine B-mode imaging of gastrocnemius and soleus muscle dynamics alone are unable to reliably estimate interactions between these muscles and their series elastic sub-tendons (*Zelik & Franz, 2017*). Thus, to gain mechanistic insight into the role of triceps surae muscle dynamics in precipitating non-uniform displacement patterns within the AT, there is a critical need for innovation in our measurement techniques; to understand the complexities of muscle and tendon behavior, you must measure muscle and tendon behavior.

Our goal was to investigate individual triceps surae (i.e., medial gastrocnemius and soleus) muscle dynamics as a determinant of non-uniform AT tissue displacement patterns during isolated contractions. We used a dual-probe ultrasound imaging technique to quantify *in vivo* gastrocnemius and soleus muscle dynamics in synchrony with localized tendon tissue displacements within their associated sub-tendons. We hypothesized that superficial versus deep differences in AT tissue displacements would be accompanied by and correlate with anatomically consistent differences in medial gastrocnemius (GAS) versus soleus (SOL) muscle shortening. We also tested the secondary hypothesis that the

relationship between AT tissue displacements and GAS versus SOL muscle shortening would vary with ankle angle, which we would interpret in the context of altered triceps surae force generating capacity (*Huijing et al., 2011*; *Kawakami, Ichinose & Fukunaga, 1998*; *Tian et al., 2012*) and tendon slack (*Herbert et al., 2011*; *Hug et al., 2013*).

## MATERIALS AND METHODS

### Subjects and protocol

We estimated that $n = 8$ subjects would have 80% power to detect ($p < 0.05$) previously reported differences between peak GAS and SOL sub-tendon displacements (i.e., 5.95 mm versus 8.49 mm) in young subjects during an isolated ankle task (*Slane & Thelen, 2015*). We recruited 12 subjects to participate and excluded three subjects during our quality control process. Specifically, we excluded subjects for whom the average frame-to-frame correlation in our speckle tracking algorithm, later described, for any one condition fell below 80%. Thus, we report data for nine subjects (age: $25.1 \pm 5.6$ years, mass: $69.8 \pm 6.9$ kg, height: $1.7 \pm 0.1$ m, four females and five males). Subjects provided written informed consent as per the UNC Biomedical Sciences Institutional Review Board (16–0379) and walked without an assistive aid, had no orthopedic disorders within the last six months nor any neurological disorder or disease, and did not have a leg prosthesis. Subjects first walked on a treadmill (Bertec Corp., Columbus, OH) for 6 min at 1.25 m/s to pre-condition their triceps surae muscle–tendon units and reach steady-state behavior (*Hawkins et al., 2009*). Subjects then performed three ramped maximum voluntary isometric contractions at each of five ankle angles (10° dorsiflexion to 30° plantarflexion in 10° increments) in a dynamometer (Biodex, Shirly, NY), with the knee flexed to replicate that near the push-off phase of walking (∼20 degrees). To elicit a symmetric loading-unloading profile, subjects started from rest and increased their plantarflexor moment until reaching their voluntary maximum at 2 s, before steadily returning to rest at 4 s. Prior to data collection, subjects briefly practiced this 4 s ramped contraction using a real-time display of their net ankle moment for positive reinforcement. We fully-randomized the ankle angles, and subjects rested at least one minute between each contraction. Subjects were barefoot throughout the experiment to allow proper placement of the ultrasound transducers.

### Measurements

We synchronized two 10 MHz linear array ultrasound transducers to simultaneously record GAS and SOL fascicle kinematics with tissue displacements in their associated tendinous structures (Fig. 1). A 60 mm Telemed Echo Blaster 128 ultrasound transducer (LV7.5/60/128Z-2, UAB Telemed, Vilnius, Lithuania) placed over the mid-belly of the GAS of subjects' right leg recorded cine B-mode images at 61 frames/s through a longitudinal cross section using an image depth of 65 mm. This Telemed transducer placement and depth also enabled imaging of the SOL in the same image plane (*Cronin et al., 2013*; *Tian et al., 2012*). Simultaneously, a 38-mm transducer (L14-5W/38, Ultrasonix Corporation, Richmond, BC) operating at 70 frames/s recorded 128 lines of ultrasound radiofrequency (RF) data from subjects' right free AT, distal to the SOL muscle–tendon junction, using an image depth of 20 mm and secured using a custom orthotic. Placement of this probe

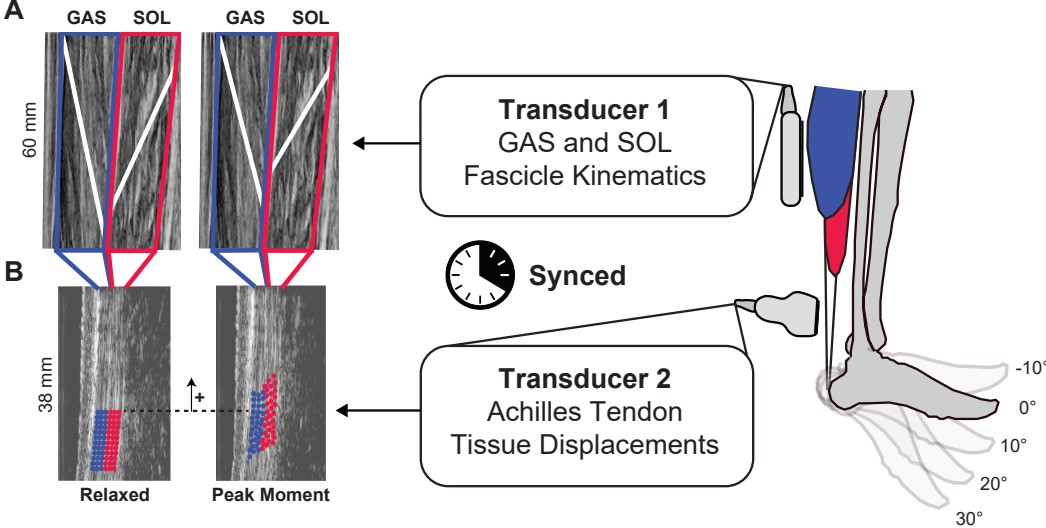

**Figure 1 Dual-probe imaging methodological approach.** We used a dual-probe ultrasound imaging approach that enables the simultaneous assessment of the medial gastrocnemius (GAS) and soleus (SOL) muscle fascicle kinematics with tissue displacements in their associated sub-tendons of the free Achilles tendon (AT). (A) Time series of fascicle lengths and pennation angles were derived from cine B-mode images. (B) A custom 2D speckle tracking algorithm estimated localized displacements of two equally sized tendon depths—superficial and deep—corresponding to tendon tissue thought to arise from GAS and SOL, respectively.

was prescribed by the fit of our custom orthotic, secured with straps just proximal to the malleoli and thus selected to replicate the placement used in prior studies (i.e., ∼6 cm proximal to the calcaneal insertion) (*Franz et al., 2015*).

Eight cameras from a 14-camera motion capture system (Motion Analysis, Corp., Santa Rosa, CA) operating at 100 Hz recorded the 3D positions of 14 retroreflective markers placed on the subjects' right lower leg and ultrasound transducers. An inverse kinematics routine (*Silder, Heiderscheit & Thelen, 2008*) estimated ankle and knee joint angles and we recorded the plantarflexor moment from the dynamometer at 1,000 Hz.

We collected binary synchronization signals from the Telemed and Ultrasonix machines at 1,000 Hz. These signals co-registered the onset of ultrasound data collected from the GAS, SOL, and free AT with the plantarflexor moment and marker trajectories. We analyzed all data between key-frames at the beginning and end of each ramped contraction, defined using a threshold of 5% peak moment. From the ultrasound data, we quantified time series of (i) GAS and SOL fascicle kinematics and (ii) tendon tissue kinematics, each interpolated to 1000 data points per trial, as described below.

## Muscle kinematics

Following best practices outlined by *Farris & Lichtwark (2016)*, the same investigator performed all muscle tracking. First, we defined a static polygon region of interest (ROI) surrounding each muscle and their aponeuroses (Fig. 1). We then defined one GAS and one SOL muscle fascicle in the mid-region of the imaged plane, considered representative
of the muscle belly, from their superficial to deep aponeurosis in the first key-frame of every trial. Open source MATLAB routines based on an affine extension to an optic flow algorithm quantified time series of GAS and SOL fascicle lengths ($L_{\text{fascicle}}$) and pennation angles ($\alpha$) (*Farris & Lichtwark, 2016*). Pennation angles corresponded to the oblique angle between the image horizontal axis and the defined fascicle of the respective muscle. Finally, to more directly place these muscle dynamics in the context of tendon tissue displacement, we combined muscle fascicle lengths and pennation angles to compute muscle length longitudinal to its line of action ($L_{\text{muscle}}$):

$$L_{\text{muscle}} = L_{\text{fascicle}} \cos\alpha.$$

## Tendon kinematics

A 2D speckle tracking algorithm estimated localized displacements of AT tissue using previously published techniques (*Chernak & Thelen, 2012*; *Chernak, Slane & Thelen, 2014*). In brief, we defined a rectangular ROI, measuring ~15 × 3 mm on a B-mode image of the free AT created from the raw RF data at the first key-frame of each trial. The ROI contained a grid of nodes with 0.83 × 0.42 mm spacing defined to encompass only tendinous tissue. A 2 mm × 1 mm kernel containing up-sampled (4×) RF data, centered at each nodal position, provided a search window over which we defined 2D normalized cross-correlation functions between successive frames. We defined localized frame-to-frame nodal displacements that maximized these 2D cross-correlations, with the cumulative displacement representing the average of forward and backward tracking results. Nodal displacements were regularized using second order polynomials (*Pan et al., 2009*). From these cumulative displacements, we quantified the average longitudinal nodal displacements originating from two equally sized tendon depths (*Szaro et al., 2009*)—superficial and deep - corresponding to tendon tissue thought to arise from GAS and SOL, respectively. This orientation represents the anatomical arrangement most prevalent in cadaveric studies (*Anson & McVay, 1971*; *Edama et al., 2015*; *Gils, Steed & Page, 1996*; *Szaro et al., 2009*). Although previous authors have acknowledged that in some anatomical observations the individual sub-tendons varied in their thickness, in the majority of the anatomical observations, the gastrocnemius sub-tendon was the same size as the soleus sub-tendon (*Anson & McVay, 1971*; *Del Buono, Chan & Maffulli, 2013*; *Doral et al., 2010*; *Gils, Steed & Page, 1996*). Here, we report these average longitudinal displacements as representing that of tissue within the corresponding GAS and SOL sub-tendons (Fig. 1). Finally, we quantified tendon non-uniformity by reporting the difference between peak tendon tissue displacements of the superficial and deep regions of the tendon.

## Statistical analysis

For each outcome measure, we took the average of the three conditions for each ankle angle. A two-way repeated measures ANOVA tested for main effects of and interactions between ankle angle and muscle–tendon unit (i.e., GAS and SOL) on peak muscle shortening and peak tendon tissue displacement using an alpha level of 0.05. When a significant main effect was found, post-hoc pairwise comparison identified the ankle angles at which GAS versus SOL differences were significant. For peak plantarflexor moment, a one-way

repeated measures ANOVA tested for significant main effects of ankle angle and Pearson's correlation coefficients assessed the relation to muscle and sub-tendon tissue kinematics. We then calculated Pearson's correlation coefficients between: (i) GAS and SOL muscle shortening and displacement in their associated regions of the AT (i.e., superficial and deep, respectively) and (ii) muscle shortening differences (i.e., GAS versus SOL) and tendon non-uniformity (i.e., superficial versus deep). As a secondary analysis, we calculated Pearson's correlation coefficients between GAS and SOL muscle shortening and displacement in their unassociated region of the AT (i.e., GAS versus deep, SOL versus superficial). Finally, we qualitatively assessed the independent contributions of fascicle length versus pennation angle on GAS and SOL muscle length changes. We plotted percent change in fascicle length versus percent change in the cosine of pennation angle (i.e., contributors to muscle length change) and noted apparent trends against the line of unity with a ratio of 1:1.

## RESULTS

Peak plantarflexor moment decreased progressively from dorsiflexion to plantarflexion across the range of angles tested ($p < 0.01$). Muscle and sub-tendon tissue kinematics were relatively independent of ankle angle; only SOL fascicle and longitudinal muscle shortening exhibited significant main effects ($p$'s $= 0.03$), with pairwise comparisons revealing smaller changes at 30° plantarflexion compared to the other conditions. However, consistent with this observation, SOL tendon displacements also tended to be smaller at 30° plantarflexion ($p = 0.064$). Across all ankle angles, the SOL shortened by an average of 78% more than the GAS during moment generation ($p < 0.01$) (Fig. 2A), due both to greater fascicle shortening and a larger increase in pennation ($p$'s $< 0.01$) (Fig. 3). However, change in fascicle length had a greater influence than change in the cosine of pennation on longitudinal GAS and SOL muscle shortening (Fig. 3).

SOL versus GAS differences at the muscle level were accompanied by, on average, 51% more displacement in the deep versus superficial region of the AT ($p < 0.01$) (Fig. 2B). Moreover, the magnitude of peak GAS and SOL muscle shortening positively correlated with peak displacements in their associated regions of the AT (GAS: $R^2 = 0.48$; SOL: $R^2 = 0.63$; $p$'s $< 0.01$) (Fig. 4A). Superficial versus deep differences in AT displacement positively correlated with anatomically consistent differences in GAS versus SOL muscle shortening ($R^2 = 0.37$, $p < 0.01$) (Fig. 4B). The strength of these latter correlations varied systematically with ankle angle, becoming stronger with increased ankle dorsiflexion (Table 1). Conversely, those correlations became non-significant at 20° ($p = 0.10$) and 30° ($p = 0.99$) plantarflexion. We also found moderate to strong correlations between GAS and SOL muscle shortening and displacement in their unassociated region of the AT ($R^2 = 0.38$ and $R^2 = 0.65$, respectively; $p$'s $< 0.05$). Finally, across all conditions, peak muscle shortening (GAS: $R^2 = 0.37$; SOL: $R^2 = 0.47$) and sub-tendon tissue displacements (GAS: $R^2 = 0.51$; SOL: $R^2 = 0.49$) significantly and positively correlated with peak plantarflexor moment.

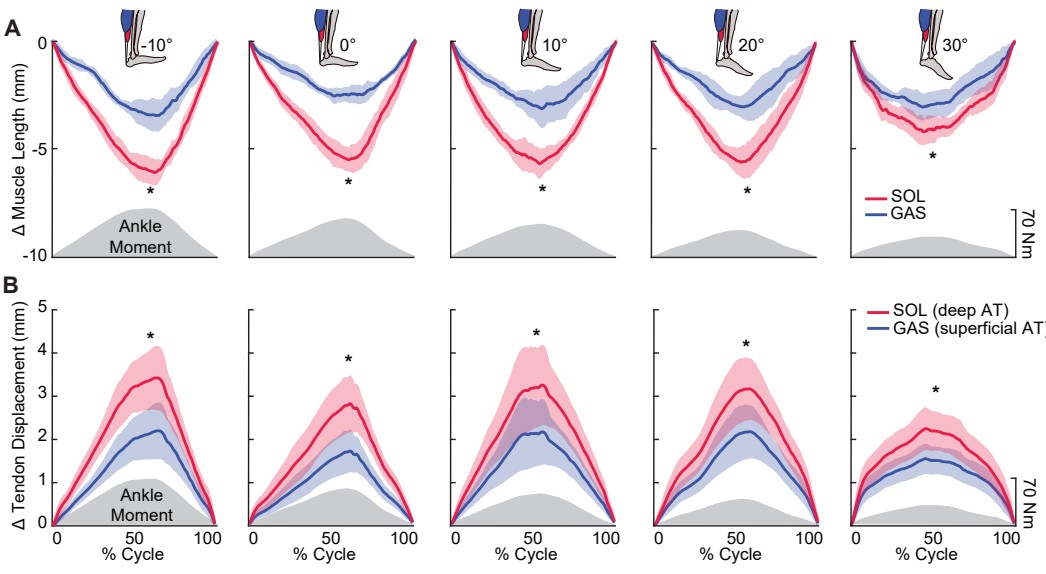

**Figure 2 Muscle shortening and sub-tendon tissue displacements across the range of ankle angles tested.** Group mean (standard error) (A) muscle shortening and (B) sub-tendon displacements (nodal displacement relative to initial position; proximal positive) across the range of ankle angles tested. Gray shaded regions show the group mean net profile during each loading-unloading cycle. Across all conditions, SOL shortened by an average of 78% more than GAS during force generation. This was accompanied by, on average, 51% more lengthening in the deep versus superficial region of the free AT. Asterisks (*) indicate statistically significant ($p < 0.05$) differences between GAS and SOL.

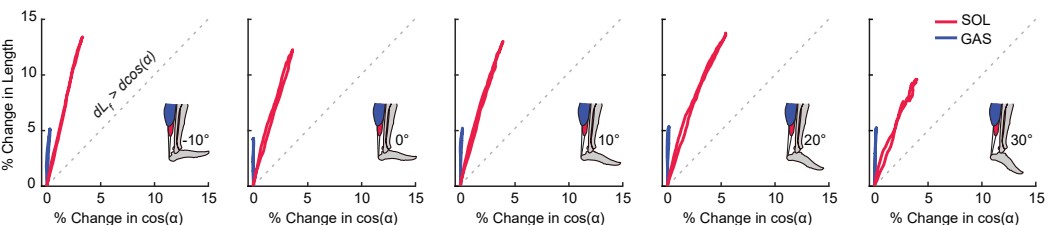

**Figure 3 Contributions of fascicle length and pennation angle to muscle shortening behavior.** Group mean percent change in fascicle length versus percent change in the cosine of pennation angle for the gastrocnemius (GAS) and soleus (SOL) across the range of ankle angles tested. The dashed line represents a line of unity (i.e., equal contributions from fascicle shortening and fascicle rotation). SOL consistently exhibited more fascicle shortening and fascicle rotation than GAS. For all conditions, fascicle shortening had a greater influence on longitudinal muscle shortening than fascicle rotation.

## DISCUSSION

We investigated the role of triceps surae muscle dynamics in precipitating non-uniform tissue displacements in the architecturally complex Achilles tendon (AT). We also used a dual-probe ultrasound imaging technique empowering the simultaneous measurement of GAS and SOL muscle fascicle dynamics together with tissue displacements within their associated sub-tendons of the AT. Our results build on dual-probe approaches applied previously to GAS quantifying muscle fascicle and muscle–tendon junction

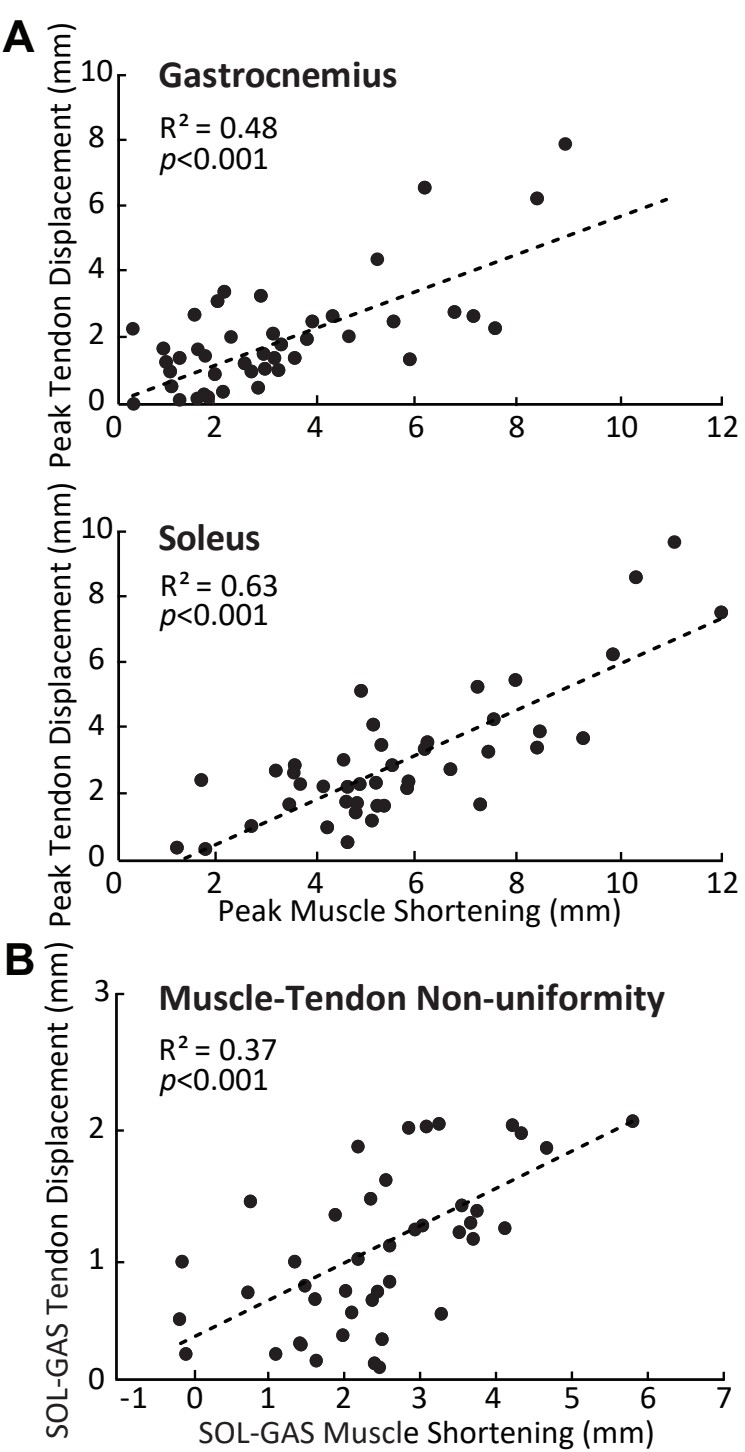

**Figure 4** **Correlations between muscle and sub-tendon tissue behavior.** (A) Correlations between peak GAS or SOL muscle shortening and displacement in their associated regions of the AT. (B) Correlation between superficial versus deep differences in AT displacement and GAS versus SOL differences in muscle shortening. All correlations are pooled across all conditions.

**Table 1** Correlations ($R^2$) and *p* values between peak muscle shortening and peak sub-tendon tissue displacements at each ankle angle.

|  |  | −10° | 0° | 10° | 20° | 30° |
|---|---|---|---|---|---|---|
| GAS | $R^2$ | **0.52** | 0.20 | **0.67** | **0.50** | 0.32 |
|  | *p*-value | **0.044** | 0.073 | **0.007** | **0.031** | 0.114 |
| SOL | $R^2$ | **0.77** | 0.38 | **0.75** | **0.76** | 0.36 |
|  | *p*-value | **0.004** | 0.075 | **0.003** | **0.002** | 0.088 |
| SOL-GAS | $R^2$ | **0.78** | **0.60** | **0.45** | 0.33 | 0.04 |
|  | *p*-value | **0.004** | **0.015** | **0.049** | 0.104 | 0.991 |

**Notes.**

Bold numbers represent significant correlations ($p < 0.05$).
GAS, Gastrocnemius; SOL, Soleus.

kinematics (*Ishikawa & Komi, 2008*; *Matijevich, Branscombe & Zelik, 2018*). Our findings largely supported our first hypothesis; independent of ankle angle, SOL shortened more than GAS during moment generation—muscle-level differences were accompanied by and correlated with anatomically consistent differences in sub-tendon tissue displacements. As we elaborate below, these findings provide empirical evidence for a mechanistic link between non-uniform displacement patterns within the human AT and the operating behavior of the triceps surae muscles.

Previous anatomical work has shown that the human free AT near our imaging location consists predominantly of superficial tendon fascicles arising from GAS and deep tendon fascicles arising from SOL (*Szaro et al., 2009*). Consistent with this architecture, the SOL muscle and its associated deep region of the AT in all cases both underwent larger kinematic changes during moment generation compared to those structures associated with GAS. Although not previously quantified simultaneously, these triceps surae muscle (*Fukashiro, Hay & Nagano, 2006*; *Hoang et al., 2007*; *Lauber, Lichtwark & Cresswell, 2014*; *Maganaris, Baltzopoulos & Sargeant, 1998*) and AT tissue (*Franz et al., 2015*) kinematics are generally consistent with those reported for a variety of activities spanning isolated contractions to functional activities. For example, we reported larger tissue displacements in the deep AT (i.e., SOL sub-tendon) than in the superficial AT (i.e., GAS sub-tendon) during the stance phase of walking, a phase in which GAS and SOL muscle fascicles exhibit different operating behavior (*Ishikawa et al., 2005a*). In their review, *Bojsen-Moller & Magnusson (2015)* summarize evidence for heterogeneous loading within the AT and at least conceptualize the potential for differential GAS and SOL activations, and thus muscle contractile dynamics, to precipitate non-uniform patterns of AT tissue displacements (*Bojsen-Moller & Magnusson, 2015*).

Despite conceptual descriptions (*Bojsen-Moller & Magnusson, 2015*) and recent model predictions (*Handsfield et al., 2017*), we lacked evidence for the role of muscle dynamics in governing non-uniform mechanical behavior within the AT. Consistent with our first hypothesis, here we found that differences between peak SOL and GAS muscle shortening positively correlated with those between peak SOL and GAS sub-tendon tissue displacement during fixed-end contractions. Coupled with anatomically consistent differences in muscle

shortening and tendon tissue displacements described above, these correlations suggest that triceps surae muscle dynamics can give rise to sliding between adjacent sub-tendons within the human AT.

We qualitatively assessed the contributions of fascicle length versus pennation angle to differences between GAS and SOL muscle contractile behavior. Fascicle length and pennation angles of the GAS and SOL, including their changes during isometric contractions, agree well with published values (*Maganaris, 2003*; *Maganaris, Baltzopoulos & Sargeant, 1998*; *Tilp et al., 2011*). In all cases, fascicle shortening contributed more than fascicle rotation to the longitudinal GAS and SOL muscle length changes. Moreover, the SOL exhibited both larger fascicle shortening and fascicle rotation than GAS. In fact, we noted very little pennation angle change in the GAS during moment generation. Thus, differences in triceps surae contractile behavior arise from a combination of larger fascicle shortening and rotation in SOL versus GAS, contributing to the non-uniform length change dynamics ultimately borne by the AT.

The strength of the correlation between tendon displacement non-uniformity and differences in GAS versus SOL muscle shortening increased with ankle dorsiflexion, which we interpret as support for our secondary hypothesis. Correlations were strongest at the most dorsiflexed ankle angle (i.e., 10°), decreasing progressively with increasing plantarflexion. This observation may reflect the effects of decreased tendon slack, decreased muscle passive tension, and/or larger peak moments conveyed by increasing ankle dorsiflexion. Indeed, muscle and tendon mechanical behavior may exhibit complex changes in response to ankle rotation (*Hug et al., 2013*). More tendon slack would imply greater tendon length change, and thus tendon tissue displacements, prior to the onset of force transmission (*Zajac, 1989*). We suspect that differences in tendon slack explain the relatively invariant tendon kinematics across the range of ankle angles tested, despite systematic changes in peak moment. Similarly, more tendon slack and/or differences between GAS and SOL tendon slack lengths may explain why these correlations became insignificant with increasing ankle plantarflexion. We also found that ankle dorsiflexion maximized peak moment, and presumably AT force transmission, consistent with prior studies (*Ackland, Lin & Pandy, 2012*; *Scovil & Ronsky, 2006*; *Zajac, 1989*). Thus, larger forces may themselves convey stronger correlations between muscle and tendon tissue dynamics. Simultaneously, or perhaps alternatively, ankle angle effects on the mechanical properties of the individual triceps surae muscles and sub-tendons themselves could influence the relation between muscle and tendon behavior. Additional study here is certainly warranted.

Although sliding between adjacent sub-tendons in the AT may convey some independence between GAS and SOL muscle–tendon actuators, myofascial force transmission through connective tissues between these muscles is also prevalent. *Oda et al. (2007)* found that an isolated stimulus to GAS elicited a similar time course and magnitude of length change in both GAS and SOL (*Oda et al., 2007*). *Kinugasa et al. (2013)* also suggested that interaponeurosis shear, elicited by adjacent muscle dynamics, influences displacements of the distal muscle–tendon junctions (*Kinugasa et al., 2013*). More recently, *Finni et al. (2017)* added that the strength of this lateral connectivity increases with muscle

activation (*Finni et al., 2017*). However, non-uniform AT displacement patterns are also present during passive ankle rotation in the absence of triceps surae muscle loading (*Arndt et al., 2012*; *Chernak, Slane & Thelen, 2014*). For this study, we interpret our findings in the context of their relevance to independent actuation; follow-up studies are necessary to better understand the relative role of lateral force transmission. Indeed, we found that SOL muscle shortening exhibited strong positive correlations with GAS sub-tendon displacement, and vice versa. One interpretation of this finding is that SOL, having by far the largest force-generating capacity of the triceps surae muscles (*Albracht, Arampatzis & Baltzopoulos, 2008*; *Ogihara et al., 2017*) coupled with having greater fascicle shortening and rotation than GAS, has a preferential influence on tendon tissue displacements in the free AT. Taking together, our results reflect independent and inter-dependent mechanical behavior between individual triceps surae muscle–tendon units. Indeed, intermuscular force transmission is documented within the triceps surae (*Huijing et al., 2011*; *Tian et al., 2012*).

There are other possible factors contributing to AT tissue non-uniformity that may act in concert with triceps surae muscle dynamics. Material property differences between adjacent sub-tendons may elicit non-uniform displacement patterns. Indeed, *Matson et al. (2012)* reported up to two-fold variations in elastic modulus between different leg tendons (*Matson et al., 2012*). In addition, architectural differences between sub-tendons (e.g., lengths and cross-sectional areas) may influence their mechanical properties (e.g., stiffness) and thus displacement patterns (*Szaro et al., 2009*). As another example, differences between sub-tendon slack lengths could also contribute. We consider the present study a first step toward an improved understanding of the relative contributions of these factors in governing triceps surae muscle–tendon behavior.

If triceps surae muscle dynamics can precipitate non-uniform Achilles tendon tissue displacements, can tendon-level changes influence muscle contractile behavior? For example, our findings may be important for understanding the functional consequences of age-associated reductions in inter-fascicle sliding within the human AT. Compared to those in young adults, we have observed more uniform AT tissue displacements in older adults during walking (*Franz & Thelen, 2015*), which may reflect collagen cross-linking and interfascicle adhesions (*Thorpe et al., 2013*). These tendon-level changes correlate with a reduced plantarflexor moment during walking in older adults, alluding to unfavorable functional consequences. Moreover, simulating a reduced capacity for inter-fascicle sliding in the AT, and thus a loss of mechanical independence between the GAS and SOL, predicts unfavorable shifts in muscle fascicle behavior during walking (*Franz & Thelen, 2016*). Our present work provides a foundation for using dual-probe imaging to gain mechanistic insight into these earlier observations.

There are several limitations of this study. First, we only report data for the medial gastrocnemius and SOL, including a generalized anatomical approximation of their associated sub-tendons (*Edama et al., 2015*; *Gils, Steed & Page, 1996*; *Szaro et al., 2009*). Second, two-dimensional imaging may not fully capture the three-dimensional behavior of the triceps surae muscles and AT. For example, the SOL muscle is comprised of anterior, posterior, medial, and lateral components that differ in architecture (*Chow et al., 2000*).

We only imaged the posteromedial SOL, which may not represent other regions. Third, muscle tracking reliability depends on meticulous fascicle determination and incumbent semi-automated tracking limitations (*Farris & Lichtwark, 2016*). To minimize these effects, one investigator performed all analyses. Fourth, the pennation angle defined by UltraTrack is not representative of a true pennation angle due to the software identifying the line of action as the horizontal axis of the image. We have also previously described the limitations of our 2D speckle tracking estimates of AT tissue displacements (*Franz et al., 2015*). We add here that cross-correlation estimates of tissue motion can be subject to spatial averaging. However, based on a previously published validation (*Chernak & Thelen, 2012*; *Chernak, Slane & Thelen, 2014*), we do not suspect this significantly influenced our findings nor interpretation. We also opted to report sub-tendon tissue displacements, an outcome we can measure with a higher level of confidence than sub-tendon elongation. Specifically, estimating sub-tendon tissue elongation relative to a motion capture estimate of calcaneal insertion can be prone to errors associated with coronal plane ankle rotation, tendon slack, and the complexities of AT curvature - errors that require further study (*Csapo et al., 2013*; *Matijevich, Branscombe & Zelik, 2018*). Fifth, we made no attempt to estimate forces transmitted through the AT, which are heterogeneous and highly complex (*Bojsen-Moller & Magnusson, 2015*). We also note that ankle and knee angles became slightly (<5°) more extended from rest to peak moment generation. The sub-tendon tissue displacements reported here likely reflect some combination of that due to muscle shortening at the proximal attachment and that due to calcaneus displacement (via ankle rotation) at the distal attachment. However, *Handsfield et al. (2017)* found a very negligible effect of retrocalcaneal tendon insertion on GAS-SOL differences in sub-tendon tissue displacement, even for 25° of ankle rotation (*Handsfield et al., 2017*). Finally, although we report anatomically consistent behavior between muscle and sub-tendons, our conclusions are based in part on correlations that cannot definitively convey causal links.

## CONCLUSIONS

We present evidence that triceps surae muscle dynamics may precipitate non-uniform displacement patterns in the architecturally complex Achilles tendon. Moreover, we used a dual-probe imaging approach to empower simultaneous assessment of muscle and tendon toward an improved mechanistic understanding of triceps surae behavior. Our findings may be important for understanding age-associated changes in AT displacement patterns, which we suspect alter muscle contractile behavior in older adults.

## ACKNOWLEDGEMENTS

We thank Ashish Khanchandani, Hannah McKenney, and Michael Browne for their assistance with data collection.

### Funding

This study was supported by a grant from the National Institutes of Health (R01AG051748). The funders had no role in study design, data collection and analysis, decision to publish, or preparation of the manuscript.

### Grant Disclosures

The following grant information was disclosed by the authors:
National Institutes of Health: R01AG051748.

### Competing Interests

The authors declare there are no competing interests.

### Author Contributions

- William H. Clark and Jason R. Franz conceived and designed the experiments, performed the experiments, analyzed the data, prepared figures and/or tables, authored or reviewed drafts of the paper, approved the final draft.

### Human Ethics

The following information was supplied relating to ethical approvals (i.e., approving body and any reference numbers):

Subjects provided written informed consent as per the UNC Biomedical Sciences Institutional Review Board (16-0379).

### Data Availability

The raw data are provided in Data S1.

### Supplemental Information

Supplemental information for this article can be found online at http://dx.doi.org/10.7717/peerj.5182#supplemental-information.

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
