# Peer review of "Do triceps surae muscle dynamics govern non-uniform Achilles tendon deformations?"

_PeerJ, doi:10.7717/peerj.5182_

## Round 0.1 · original submission · Major Revisions

The three reviewers find plenty of value in this study and I concur. However, they all have constructive critiques, some of them fairly major and requiring at least a bit of extra analysis. I agree with reviewer 2 that self-citations etc must be minimized as this is a very active area of inquiry where broad citations could easily be achieved. Nonetheless this seems like it will be a publishable study with some revision. We thank you for submitting this interesting paper and look forward to the revised MS, which will be re-reviewed.

·

Basic reporting

no comment

Experimental design

this is a very clever experimental design.

Validity of the findings

no comment

Additional comments

I would like to thank the editor and the authors for allowing me to review this very interesting manuscript. In this study, the authors tested the hypothesis that the non-uniform Achilles tendon displacement behavior could be governed by contractile dynamics of the gastrocnemius and soleus muscles. To test this hypothesis, the authors employed a very elaborate and novel experimental design to quantify the in-vivo behavior of the gastrocnemius/soleus simultaneously with Achilles tendon displacement patterns (using two time-synched ultrasound probes) during a series of isometric tasks. The final results/data provided good support that the independent contractile behavior of gastrocnemius and soleus muscles are related to the non-uniform displacement profile of the Achilles tendon. Overall, this is a very well-written manuscript and I believe that this manuscript makes an important contribution to the field. I have a few questions and comments for the next submission (ordered by the importance of the issue, see below).

1. It is not entirely clear from the Methods section how exactly the superficial and deep areas of the tendon were determined from the ultrasound image (from ‘Transducer 2’ on Figure 1). This distinction seems very important for this experiment, since the sub-tendon areas are assumed to be representative of the portions that are attributed to the gastrocnemius and soleus. This information may be described in a previous publication, but more detail in the Methods section regarding the identification of deep and superficial tendon may be helpful.

2. From the two ultrasound data, it appears that the main dependent variables are focused on the shortening of the muscle fascicle or the displacement of the Achilles tendon along the longitudinal axis of the ultrasound probe. I wonder if there was any attempt to align the longitudinal axes of the two transducers, such that the displacement values from one transducer would match one-to-one with the displacement from the other transducer. My guess is that the ultrasound probe orientation from both transducers needed to be adjusted on a subject-by-subject basis to get a clear image of the muscle fascicle or tendon. If this was the case, was Transducer 1 ever obliquely oriented relative to Transducer 2? Was the orientation of the two ultrasound probes ever recorded? Perhaps, a 10-20 degree rotation (between transducers) may not be such a big deal, but 30-40 degree rotation could start to affect the data when the goal is to relate the displacement values from one transducer to another.

3. During an isometric contraction, I would expect that the length of the entire muscle-tendon unit would remain the same. That is, the shortening of the muscle fascicles would be offset by the elongation of the tendon. From Figure 2, it seems like the amount of gastrocnemius muscle shortening comes close to offsetting the displacement of the tendon (at least, to the naked eye). Though, for the soleus muscle, the amount of shortening appears more than the tendon displacement. Do you think this could be due to an overestimation of the muscle fascicle shortening, or underestimation of the tendon displacement (or some combination of the two)? I don’t necessarily expect the authors to be able to fully address/answer this question, but I would appreciate any thoughts on this.

4. The order of Figures 3 and 4 may have been switched in the PDF document that I received.

Reviewer 2 ·

Basic reporting

Clear and unambiguous text: Generally speaking, this paper is well-written. However, at times, the highly specific and overly flowery language can be distracting. It is advised that the authors consider shortening some of the longer/less clear sentences (e.g. Lines 64-68, Lines 87-90, Line 121) to improve clarity.

Literature references: The authors do a nice job introducing the reader to the subject, and explaining prior work, however some missed references are troubling. The authors seemingly preferentially cite their own work, and are missing key references that could potentially call into question some of their claims. For example, a number of prior studies have collected synchronous ultrasound data from two probes to investigate Achilles tendon mechanics, of which the authors only mention one, late in the discussion, after claiming in the abstract, introduction and discussion that they are introducing a ‘novel’ technique. They should remove the word ‘novel’ as it is misleading to anyone unfamiliar with the literature, and is inappropriate here.

The authors also reference the work of Szaro which shows the sub-tendons of the Achilles, but it is not properly described. They claim this work shows that the deep tendon is composed of soleus sub-tendons whereas Fig. 6 in the paper suggests the lateral gastrocnemius is also contributing to the make-up of this part of the tendon. Further, Edama 2015 demonstrates the variability of Achilles tendon anatomy which is also not cited. In addition, the pre-conditioning presumably comes from the work of Hawkins et al., but again is not cited. Some of these key misses in the literature are troubling.

Professional article structure: The article is very well-organized and complete. The authors should be careful with words like “discovered” (Line 90); “observed” would be a better and more professional choice.

Self-contained with relevant results to hypotheses: Generally this is well-done, but in the abstract the hypothesis should be stated earlier on.

Experimental design

Fits with Aims and Scope: Very well.

Research question well defined, relevant and meaningful. Yes.

Rigorous investigation performed to a high technical and ethical standard. Yes.

Methods described with sufficient detail and information to replicate. Yes.

Validity of the findings

Data is robust, statistically sound & controlled: The authors should justify their sample size, and clarify if any subjects had prior injuries specific to the triceps surae complex. Tendinopathy is known to lead to long-term changes in tendon structure, which could alter sub-tendon capacity for sliding.

Conclusions are well-stated: The major concern with this paper in terms of methodology is the claim that the deep tissue corresponds to the soleus sub-tendon, and the superficial tissue corresponds with the medial gastrocnemius sub-tendon. As mentioned briefly above, this does not account for the twist in the Achilles and doesn’t even seem to be supported by the paper they are referencing. In addition, the Achilles tendon shows enormous subject-to-subject variability (O’Brien 1984) which detracts from the claims being made by the authors.

Additional comments

The authors present very interesting data to test an exciting hypothesis currently being tested in the field. The paper is very well-organized, with solid methodology, and interesting discussion. Major concerns are addressed above. Below, specific line-by-line feedback may be found below.

Abstract, Line 45. The authors express that their hypothesis was supported prior to identifying their hypothesis. The hypothesis should be described earlier.

Line 67. Shouldn’t the later-cited 2014 paper be cited here in addition?

Line 82. This line suggests that “direct ultrasound imaging techniques” were developed after the papers cited from 2005 and 2013, whereas the seminal work of Maganaris and Paul on measuring in vivo mechanical properties of tendon was published in 1999.

Line 107. Muscle and tendon have also been shown to exhibit altered mechanical properties at different ankle angles. How might this influence your hypothesis?

Line 118. How was this highly specific approach controlled? Did subjects get practice?

Line 121. The structure of this sentence is awkward.

Line 137. What I meant by “manually adjusted in transverse rotation”?

Line 144. How were ultrasound data synchronized at 1000 Hz if the frame rate was only 70 frames/s and 61 frames/s? How did

Line 178. Did the authors do any quality control on their data? Or were all collected trials included in analysis?

Line 197. The authors should clarify what is meant that they “qualitatively assessed” results.

It would be interesting if the authors could discuss how their results fit into the recent publication by Matijevich et al., who observed unexpected shortening of the Achilles tendon during ankle plantarflexion. Isn’t the assumption of a series connection between muscles and sub-tendons called into question by this recent publication?

Line 237. This line seems to contrast with Fig. 6 in the cited paper which clearly identifies the lateral gastrocnemius in the deep portion of the tendon. Edama 2015 also suggests that there are substantial variations in human Achilles tendon anatomy, with it appearing quite common that the deep tissue observed could arise in the lateral gastrocnemius.

Lines 272-283. How might the observations of changing tissue mechanical properties in both the gastrocnemius and Achilles tendon at different ankle angles relate to these findings?

Figure 1 is very nice and shows great clarity, however, again the authors should refrain from terming this approach as novel. Also, what about the twist in the Achilles? Based on the referenced Szaro paper, wouldn’t the deep portion of the tendon be part of the lateral gastrocnemius sub-tendon?

Reviewer 3 ·

Basic reporting

Throughout the manuscript, please define GAS and SOL and then use abbreviation consistently. At present, there are some abbreviations and some full spellings and the authors tend to switch back and forth throughout.

The figures were not in the correct order within the version I received to review. It is unclear whether the authors uploaded incorrectly or this was a journal formatting issue. Please check on resubmission.

Experimental design

No comment

Validity of the findings

No comment

Additional comments

Simultaneous analysis of muscle and tendon behavior is imperative to understanding the functional behavior and force-generating ability of muscle-tendon units. This study introduced a novel dual-probe ultrasound imaging approach to simultaneously measure muscle fascicle behavior and tendinous displacements in the triceps surae muscle-tendon unit. The premise for the study is based on previous work from the senior author that suggests non-uniform tissue displacements within the Achilles tendon is a functionally important mechanism for human movement. The results display differential behavior between the medial gastrocnemius, and its associated subtendon and the soleus and its associated subtendon. The manuscript is well-written and the experiments were carefully executed. As discussed below, you will see that I have a few comments and suggestions that I would like to see addressed.

Major Comments:

On Line 76, the authors suggest (and provide reference) that differential forces are likely to be the best candidate to explain non-uniform AT strains. However they do not examine the influence of force in their results. They highlight (Line 341) that the forces transmitted via the AT are ‘heterogeneous and highly complex.’ Yet the authors measure ankle moment (and could presumably estimate moment arm given the authors previous expertise). Using a total ankle moment (or force, if subject specific moment arms are accessible) together with the estimated relative PCSAs of the SOL and GAS in relation to the total plantarflexor force-generating ability (perhaps assuming similar activation levels), the authors should be able to determine the effect of force on their results. At minimum, I would like to see the effect of torque from the dynamometer included within their repeated measures ANOVA.


The authors find an R2 of 0.63 for SOL muscle shortening with peak SOL tendon displacement, but then find an R2 of 0.65 for SOL muscle shortening with peak GAS tendon displacement. The stronger correlation between soleus fascicle behavior and gastroc sub-tendon displacement seems to be an important finding that is not adequately discussed within the manuscript. This finding suggests, in part, that the anatomical associations between the MTUs of the triceps surae group restrict independent actuation. The authors hypothesize that superficial versus deep differences in AT tissue displacements would be accompanied by, and correlate with, anatomically consistent difference in GAS vs SOL muscle shortening. In the discussion (Line 231-234) they suggest that the findings fully support this hypothesis. Their results do indeed support this BUT also suggest that muscle-level differences are even more strongly associated with sub-tendons that are NOT anatomically consistent. I would like to see this addressed.

Discussion Paragraph Line 261-268 – the rationale for assessing the relative contributions of muscle length change versus pennation angle change remain unclear. Further, the paragraph within the discussion should include and compare these findings to other studies that have measured simultaneous changes in fascicle length and pennation angle.

Minor Comments:

Abstract – please highlight that you are measuring features of the medial gastrocnemius specifically

Line 60 – How much is largely? Can you provide the amount of total lower limb power provided by the plantarflexors/ankle.

Line 97 – please state that this is medial gastrocnemius

Line 114 – please provide evidence for why pre-conditioning the MTUs prior to tests is necessary. It is currently unclear why this important?

Line 115 – Include ‘maximum’ before isometric voluntary contractions

Line 147 – it is unclear why a threshold of 5% peak moment was necessary.

Line 154 – was this fascicle chosen to be in the mid-region of the muscle belly? Please clarify

Line 130 – enable should be enabled to be consistent with past tense throughout methods.

Line 203 – please test this effect of moment or force (rather than just ankle angle). See major comment 2.

Line 214 – ‘moreover’ seems unnecessary here.

Results section – the figures should be cited in numerical order (ie Fig 1, 2, 3, 4) but currently Fig 4 comes before Fig 3.

Line 269 – sentence should start with ‘The’.

Line 275-276 – the authors suggest that there are relatively invariant tendon kinematics, however Figure 2 suggests that the tendon lengthens by approx. 30 % more at -10 deg as compared to +30 deg. The is quite a large difference.

Line 278 – the authors mention subject variability as a possible reason for why the correlations became insignificant with increasing ankle PF. Presumably they have looked at this data, and can definitively state whether the differences between subjects were greater at these specific angles. Could it be that at these ankles, some subjects tended to lift their heel off the foot plate more than others, and this was a major contributor to the subject differences?

Line 319 – the authors suggest that these dual probe methods could be employed during more dynamic tasks (like walking) to help provide mechanistic insight into earlier findings (ageing locomotion). Is this true? Would it be feasible to have the 2 probes secured during a more dynamic task, like walking?

Line 330- It would be helpful to include how the effect of a horizontal registered pennation angle (due to Ultratrak methods) may influence the results. For example, were the aponeuroses typically parallel with the imaging plane? Does this even matter in the context of your results where you calculate % changes in pennation rather than angular values?

Discussion– can the authors suggest how their findings might compare to tendon tracking based on the position of the MTJ and the fascicle, rather than the sub-tendon tracking algorithm used here and the B-mode fascicle tracking. Would they expect similar correlations and findings?

---

## Round 0.2 · accepted · Accept

Congratulations, the reviewers are all pleased now! One tiny correction suggestion. Also, please ensure that as much of your data are made publicly available (e.g. in a repository or the Supp Info), within reason for anonymity of human subjects etc., to satisfy the principles of Open Science. Thanks for submitting this nice paper!

Reviewer 3 ·

Basic reporting

NA

Experimental design

NA

Validity of the findings

NA

Additional comments

In line 32 of the introduction, the gastrocnemius and soleus are first stated, but then not abbreviated and defined until the end of the intro. This is not the case for the Achilles tendon (AT) which is defined in the first line of the intro, with the abbreviation used following this. I ask the authors to update this to be consistent.